# Understanding and Improving Lexical Choice in Non-Autoregressive Translation

**Liang Ding[1]\*, Longyue Wang[2], Xuebo Liu[3], Derek F. Wong[3], Dacheng Tao[1] & Zhaopeng Tu[2]**
[1]The University of Sydney     [2]Tencent AI Lab     [3]University of Macau
{ldin3097,dacheng.tao}@sydney.edu.au, nlp2ct.xuebo@gmail.com,
{vinnylywang,zptu}@tencent.com, derekfw@um.edu.com

## Abstract

Knowledge distillation (KD) is essential for training non-autoregressive translation (NAT) models by reducing the complexity of the raw data with an autoregressive teacher model. In this study, we empirically show that as a side effect of this training, the lexical choice errors on low-frequency words are propagated to the NAT model from the teacher model. To alleviate this problem, we propose to expose the raw data to NAT models to restore the useful information of low-frequency words, which are missed in the distilled data. To this end, we introduce an extra Kullback-Leibler divergence term derived by comparing the lexical choice of NAT model and that embedded in the raw data. Experimental results across language pairs and model architectures demonstrate the effectiveness and universality of the proposed approach. Extensive analyses confirm our claim that our approach improves performance by reducing the lexical choice errors on low-frequency words. Encouragingly, our approach pushes the SOTA NAT performance on the WMT14 English-German and WMT16 Romanian-English datasets up to 27.8 and 33.8 BLEU points, respectively.

## 1 Introduction

When translating a word, translation models need to spend a substantial amount of its capacity in disambiguating its sense in the source language and choose a lexeme in the target language which adequately express its meaning (Choi et al., 2017; Tamchyna, 2017). However, neural machine translation (NMT) has a severe problem on lexical choice, since it usually has mistranslation errors on low-frequency words (Koehn & Knowles, 2017; Nguyen & Chiang, 2018; Gu et al., 2020).

In recent years, there has been a growing interest in non-autoregressive translation (NAT, Gu et al., 2018), which improves decoding efficiency by predicting all tokens independently and simultaneously. Well-performed NAT models are generally trained on synthetic data distilled by autoregressive translation (AT) teachers instead of the raw training data (Figure 1(a)) (Stern et al., 2019; Lee et al., 2018; Ghazvininejad et al., 2019; Gu et al., 2019; Hao et al., 2021). Recent studies have revealed that knowledge distillation (KD) reduces the modes (i.e. multiple lexical choices for a source word) in the raw data by re-weighting the training examples (Furlanello et al., 2018; Tang et al., 2020), which lowers the intrinsic uncertainty (Ott et al., 2018) and learning difficulty for NAT (Zhou et al., 2020; Ren et al., 2020). However, the side effect of KD has not been fully studied. In this work,

| | SRC | 今天 纽马 基特 的 跑道 湿软 。 |
|---|---|---|
| | RAW-TGT | The going at **Newmarket** is soft ... |
| | KD-TGT | Today, *Newmargot*'s runway is soft ... |
| | SRC | 纽马 基特 赛马 总是 吸引 ... |
| | RAW-TGT | The **Newmarket** stakes is always ... |
| | KD-TGT | The *Newmarquette* races always ... |
| | SRC | 在 纽马 基特 3 时 45 分 那场 中 , 我 ... |
| | RAW-TGT | I've ... in the 3.45 at **Newmarket**. |
| | KD-TGT | I ... at 3:45 a.m. in *Newmarquite*. |

Table 1: All samples that contain the source word "纽马 基特" in raw and distilled training corpora, which are different in target sides (RAW-TGT vs. KD-TGT).

---

\*Work was done when Liang Ding and Xuebo Liu were interning at Tencent AI Lab.

we investigate this problem from the perspective of lexical choice, which is at the core of machine translation.

We argue that the lexical choice errors of AT teacher can be propagated to the NAT model via the distilled training data. To verify this hypothesis, we qualitatively compare raw and distilled training corpora. Table 1 lists all samples whose source sentences contain the place name "纽马基特". In the raw corpus ("RAW-TGT"), this low-frequency word totally occurs three times and corresponds to correct translation "Newmarket". However, in the KD corpus ("KD-TGT"), the word is incorrectly translated into a person name "Newmargot" (Margot Robbie is an Australian actress) or organization name "Newmarquette" (Marquette is an university in Wisconsin) or even invalid one "Newmarquite".

Motivated by this finding, we explore NAT from the *lexical choice perspective*. We first validate our hypothesis by analyzing the lexical choice behaviors of NAT models (§3). Concretely, we propose a new metric AoLC (*accuracy of lexical choice*) to evaluate the lexical translation accuracy of a given NAT model. Experimental results across different language pairs show that NAT models trained on distilled data have higher accuracy of global lexical translation (AoLC↑), which results in better sequence generation. However, fine-grained analyses revealed that although KD improves the accuracy on high-frequency tokens, it meanwhile harms performance on low-frequency ones (*Low freq.* AoLC↓). And with the improvement of teacher models, this issue becomes more severe. We conclude that the lexical choice of the low-frequency tokens is a typical kind of *lost information* when using knowledge distillation from AT model.

In order to rejuvenate this lost information in raw data, we propose to expose the raw data to the training of NAT models, which augments NAT models the ability to learn the lost knowledge by themselves. Specifically, we propose two bi-lingual lexical-level data-dependent priors (*Word Alignment Distribution* and *Self-Distilled Distribution*) extracted from raw data, which is integrated into NAT training via Kullback-Leibler divergence. Both approaches expose the lexical knowledge in the raw data to NAT, which makes it learn to restore the useful information of low-frequency words to accomplish the translation.

We validated our approach on several datasets that widely used in previous studies (i.e. WMT14 En-De, WMT16 Ro-En, WMT17 Zh-En, and WAT17 Ja-En) and model architectures (i.e. MaskPredict (Ghazvininejad et al., 2019) and Levenshtein Transformer (Gu et al., 2019)). Experimental results show that the proposed method consistently improve translation performance over the standard NAT models across languages and advanced NAT architectures. The improvements come from the better lexical translation accuracy (low-frequency tokens in particular) of NAT models (AoLC↑), which leads to less mis-translations and low-frequency words prediction errors. The main contributions of this work are:

- Our study reveals the side effect of NAT models' knowledge distillation on low-frequency lexicons, which makes the standard NAT training on the distilled data sub-optimal.

- We demonstrate the necessity of letting NAT models learn to distill lexical choices from the raw data by themselves.

- We propose an simple yet effective approach to accomplish this goal[1], which are robustly applicable to several model architectures and language pairs.

## 2 Preliminaries

### 2.1 Non-Autoregressive Translation

The idea of NAT has been pioneered by Gu et al. (2018), which enables the inference process goes in parallel. Different from AT models that generate each target word conditioned on previously generated ones, NAT models break the autoregressive factorization and produce target words in parallel. Given a source sentence $\mathbf{x}$, the probability of generating its target sentence $\mathbf{y}$ with length $T$ is calculated as:

$$p(\mathbf{y}|\mathbf{x}) = p_L(T|\mathbf{x}; \theta) \prod_{t=1}^{T} p(\mathbf{y}_t|\mathbf{x}; \theta) \tag{1}$$

---

[1]Code is available at: `https://github.com/alphadl/LCNAT`

where $p_L(\cdot)$ is a separate conditional distribution to predict the length of target sequence. During training, the negative loglikelihood loss function of NAT is accordingly $\mathcal{L}_{\mathrm{NAT}}(\theta) = -\log p(\mathbf{y}|\mathbf{x})$. To bridge the performance gap between NAT and AT models, a variety approaches have been proposed, such as multi-turn refinement mechanism (Lee et al., 2018; Ghazvininejad et al., 2019; Gu et al., 2019; Kasai et al., 2020), rescoring with AT models (Wei et al., 2019; Ma et al., 2019; Sun et al., 2019), adding auxiliary signals to improve model capacity (Wang et al., 2019; Ran et al., 2019; Guo et al., 2019; Ding et al., 2020), and advanced training objective (Wei et al., 2019; Shao et al., 2019; Ma et al., 2020). Our work is complementary to theirs: while they focus on improving NAT models trained on the distilled data, we refine the NAT models by exploiting the knowledge in the raw data.

**Sentence-Level Knowledge Distillation** NAT models suffer from the *multimodality problem*, in which the conditional independence assumption prevents a model from properly capturing the highly multimodal distribution of target translations. For example, one English source sentence "Thank you." can be accurately translated into German as any one of "Danke.", "Danke schön." or "Vielen Dank.", all of which occur in the training data.

To alleviate this problem, Gu et al. (2018) applied sequence-level KD (Kim & Rush, 2016) to construct a synthetic corpus, whose target sentences are generated by an AT model trained on the raw data, as shown in Figure 1(a). The NAT model is only trained on distilled data with lower modes, which makes it easily acquire more deterministic knowledge (e.g. one lexical choice for each source word). While separating KD and model training makes the pipeline simple and efficient, it has one potential threat: *the re-weighted samples distilled with AT model may have lost some important information.* Lee et al. (2020) show that distillation benefits the sequence generation but harms the density estimation. In this study, we exploit to bridge this gap by exposing the raw data to the training of NAT models, as shown in Figure 1(b).

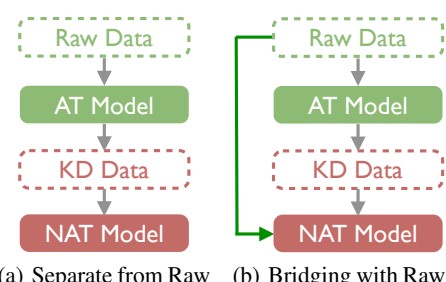

(a) Separate from Raw   (b) Bridging with Raw

Figure 1: Comparison of existing two-step and our proposed NAT training scheme.

## 2.2 EXPERIMENTAL SETUP

**Datasets** Experiments were conducted on four widely-used translation datasets: WMT14 English-German (En-De, Vaswani et al. 2017), WMT16 Romanian-English (Ro-En, Gu et al. 2018), WMT17 Chinese-English (Zh-En, Hassan et al. 2018), and WAT17 Japanese-English (Ja-En, Morishita et al. 2017), which consist of 4.5M, 0.6M, 20M, and 2M sentence pairs, respectively. We use the same validation and test datasets with previous works for fair comparison. To avoid unknown words, we preprocessed data via BPE (Sennrich et al., 2016) with 32K merge operations. The GIZA++ (Och & Ney, 2003) was employed to build word alignments for the training datasets. We evaluated the translation quality with BLEU (Papineni et al., 2002).

**NAT Models** We validated our research hypotheses on two SOTA NAT models:

- *MaskPredict* (MaskT, Ghazvininejad et al. 2019) that uses the conditional mask LM (Devlin et al., 2019) to iteratively generate the target sequence from the masked input. We followed its optimal settings to keep the iteration number be 10 and length beam be 5, respectively.

- *Levenshtein Transformer* (LevT, Gu et al. 2019) that introduces three steps: deletion, placeholder prediction and token prediction. The decoding iterations in LevT adaptively depends on certain conditions.

For regularization, we tune the dropout rate from [0.1, 0.2, 0.3] based on validation performance in each direction, and apply weight decay with 0.01 and label smoothing with $\epsilon = 0.1$. We train batches of approximately 128K tokens using Adam (Kingma & Ba, 2015). The learning rate warms up to $5 \times 10^{-4}$ in the first 10K steps, and then decays with the inverse square-root schedule. We followed the common practices (Ghazvininejad et al., 2019; Kasai et al., 2020) to evaluate the translation performance on an ensemble of top 5 checkpoints to avoid stochasticity.

| Dataset | En-De | | | Zh-En | | | Ja-En | | |
|---------|-----|------|------|-----|------|------|-----|------|------|
| | **CoD** | **AoLC** | **BLEU** | **CoD** | **AoLC** | **BLEU** | **CoD** | **AoLC** | **BLEU** |
| **Raw** | 3.53 | 74.3 | 24.6 | 5.11 | 68.5 | 22.6 | 3.92 | 73.1 | 27.8 |
| **KD (BASE)** | 1.85 | 75.5 | 26.5 | 3.23 | 71.8 | 23.6 | 2.80 | 74.7 | 28.4 |
| **KD (BIG)** | 1.77 | 76.3 | 27.0 | 3.01 | 72.7 | 24.2 | 2.47 | 75.3 | 28.9 |

Table 2: Results of different metrics on the MaskT model trained on different datasets. "KD (X)" denotes the distilled data produced by the AT model with X setting. "CoD" denotes the complexity of data metric proposed by Zhou et al. (2020), and "AoLC" is our proposed metric to evaluate the accuracy of lexical choice in NAT models.

**AT Teachers** We closely followed previous works on NAT to apply sequence-level knowledge distillation (Kim & Rush, 2016) to reduce the modes of the training data. More precisely, to assess the effectiveness of our method under different of AT teachers, we trained three kinds of Transformer (Vaswani et al., 2017) models, including Transformer-BASE, Transformer-BIG and Transformer-STRONG. The main results employ LARGE for all directions except Ro-En, which is distilled by BASE. The architectures of Transformer-BIG and Transformer-STRONG are unchanged, but STRONG utilizes a large batch (458K tokens) training strategy.

## 3 UNDERSTANDING LEXICAL CHOICE IN NAT MODELS

### 3.1 EVALUATING LEXICAL CHOICE OF NAT MODELS

Recently, Zhou et al. (2020) argue that knowledge distillation is necessary for the uncertain nature of the machine translation task. Accordingly, they propose a metric to estimate the complexity of the data ($CoD$), which is driven from an external word alignment model. They reveal that the distilled data is indeed less complex, which facilitates easier training for the NAT model. Inspired by this, we propose a metric to measure the lexical level accuracy of model predictions.

**Accuracy of Lexical Choice (AoLC)** evaluates the accuracy of target lexicon chosen by a trained NAT model $M$ for each source word. Specifically, the model $M$ takes a source word $f$ as the input, and produce a hypothesis candidate list with their corresponding word confidence:

$$\mathbf{P}_f^M = \{P^M(e_1|f), \ldots, P^M(e_{|\mathbf{V}_{trg}|}|f)\} \tag{2}$$

where $\mathbf{V}_{trg}$ is the target side vocabularies over whole corpus. The AoLC score is calculated by averaging the probability of the gold target word $e_f$ of each source word $f$:

$$AoLC = \frac{\sum_{f \in \mathbf{V}_{src}^{test}} P^M(e_f|f)}{|\mathbf{V}_{src}^{test}|} \tag{3}$$

where $\mathbf{V}_{src}^{test}$ is the set of source side tokens in test set. Each gold word $e_f$ is chosen with the help of the word alignment model $P_f^A$. The chosen procedure is as follows: Step 1) collecting the references of the source sentences that contains source word $f$, and generating the target side word bag $\mathbb{B}_f$ with these references. Step 2) Descending $P_f^A$ in terms of alignment probabilities and looking up the word that first appears in $\mathbb{B}_f$ as the gold word until the $\mathbb{B}_f$ is traversed. Step 3) If the gold word is still not found, let the word with the highest alignment probability in $P_f^A$ as the gold word. Generally, higher accuracy of lexical translation represents more confident of the predictions. We discuss the reliability of word alignment-based AoLC in Appendix A.1.

### 3.2 GLOBAL EFFECT OF KNOWLEDGE DISTILLATION ON LEXICAL CHOICE

In this section, we analyze the lexical choice behaviors of NAT models with our proposed AoLC. In particular, We evaluated three MaskT models, which are respectively trained on the raw data, AT-BASE and AT-BIG distilled data. We compared the AoLC with other two metrics (i.e. BLEU and CoD) on three different datasets (i.e. En-De, Zh-En and Ja-En). As shown in Table 2, KD is able to

improve translation quality of NAT models (BLEU: KD(BIG) >KD(BASE) >Raw) by increasing the lexical choice accuracy of data (AoLC: KD(BIG) >KD(BASE) >Raw). As expected, NAT models trained on more deterministic data (CoD↓) have lower lexical choice errors (AoLC↑) *globally*, resulting in better model generation performance (BLEU↑).

### 3.3 DISCREPANCY BETWEEN HIGH- AND LOW-FREQUENCY WORDS ON LEXICAL CHOICE

To better understand more detailed lexical change within data caused by distillation, we break down the lexicons to three categories in terms of frequency. And we revisit it from two angles: training data and translated data.

We first visualize the changing of training data when adopting KD in terms of words frequency density. As shown in Figure 2, we find that the kurtosis of KD data distribution is higher than that of raw, which becomes more significant when adopting stronger teacher. The side effect is obvious, that is, the original high- / low-frequency words become more / fewer, making the distribution of training data more imbalance and skewed, which is problematic in data mining field (Chawla et al., 2004). This discrepancy may erode the *translation performance of low-frequency words* and *generalization performance on other domains*. Here we focus on low-frequency words, and generalization performance degradation will be exploited in future work.

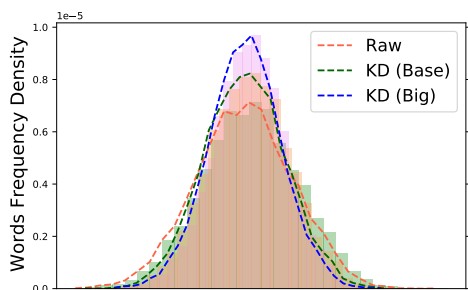

Figure 2: Comparison of the token frequency density (w.r.t the sampled tokens' probability distribution) between *Raw*, *KD (Base)* and *KD (Big)* WMT14 En-De training data.

In order to understand the detailed change during inference, we then analyze the lexical accuracy with different frequencies in the test set. We make the comprehensive comparison cross languages based on our proposed AoLC. As shown in Figure 3, as the teacher model becomes better, i.e. KD(base)→KD(big), the lexical choice of high-frequency words becomes significantly more accurate (AoLC ↑) while that of low-frequency words becomes worse (AoLC ↓). Through fine-grained analysis, we uncover this interesting discrepancy between high- and low- frequency words. The same phenomena (lexical choice errors on low-frequency words propagated from teacher model) also can be found in general cases, e.g. distillation when training smaller AT models. Details can be found in Appendix A.2. To keep the accuracy of high-frequency words and compensate for the imbalanced low-frequency words caused by KD, we present a simple yet effective approach below.

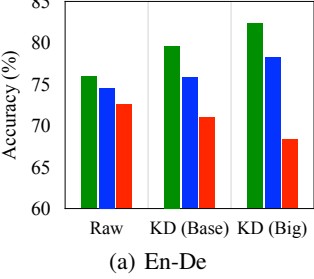
(a) En-De

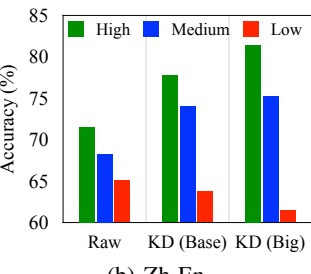
(b) Zh-En

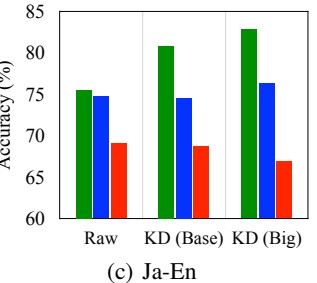
(c) Ja-En

Figure 3: Accuracy of lexical choice (AoLC) for source words of different frequency.

# 4 IMPROVING LEXICAL CHOICE IN NAT MODELS

## 4.1 METHODOLOGY

Our goal is to augment NAT models to learn needed lexical choices from the raw data to achieve better performance. To this end, we introduce an extra bilingual data-dependent prior objective to augment the current NAT models to distill the required lexical choices from the raw data. Specifically, we use Kullback-Leibler divergence to guide the probability distribution of model predictions $P^M(e|\mathbf{f})$ to match the prior probability distributions $Q(\cdot)$:

$$\mathcal{L}_{prior} = -\sum_{e \in \mathbf{e}} \text{KL}\big(Q(e|\mathbf{f}) \,||\, P^M(e|\mathbf{f})\big) \tag{4}$$

where $\mathbf{f}$ is the source sentence, and $\mathbf{e}$ is the target sentence. The bilingual prior distribution $Q(\cdot)$ is derived from the raw data, which is independent of the model $M$ and will be described later. The final objective for training the NAT model becomes:

$$\mathcal{L} = (1 - \lambda)\mathcal{L}_{NAT} + \lambda\mathcal{L}_{prior} \tag{5}$$

in which the imitation rate $\lambda$ follows the logarithmic decay function:

$$\lambda(i) = \begin{cases} \frac{log(\text{I}/(2(i+1)))}{log(\text{I}/2)} & i \leq \text{I}/2 \\ 0 & \text{others} \end{cases} \tag{6}$$

where $i$ is the current step, I is the total training step for distilled data. Accordingly, the NAT model is merely fed with the priori knowledge derived from the raw data at beginning. Along with training, the supervision signal of the prior information is getting weaker while that of the distilled data gradually prevails in the training objective. We run all models for 300K steps to ensure adequate training, thus the bilingual prior distributions will be exposed at the first 150K steps.

**Choices of Prior Distribution** $Q(\cdot)$     The goal of the prior objective is to guide the NAT models to learn to distill the lexical choices itself from the raw data. For each target word $e$, we use the external word alignment to select the source word $f$ with the maximum alignment probability, and $Q(\cdot)$ is rewritten as:

$$Q(e|\mathbf{f}) = Q(e|f) \tag{7}$$

Specifically, we use two types of bilingual prior distributions:

- *Word Alignment Distribution (WAD)* is the distribution derived from the external word alignment $\mathbf{P}_f^D = \{P^D(e_1|f), \ldots, P^D(e_N|f)\}$ where $\{e_1, \ldots, e_N\}$ are the set of target words aligned to the source word in the training data. We follow Hinton et al. (2015) to use the softmax temperature mechanism to map $\mathbf{P}_f^D$ over the whole target vocabulary:

$$Q(e|f) = \hat{\mathbf{P}}_f^D = \frac{exp(\mathbf{P}_f^D/\tau)}{\sum_{V_{tgt}} exp(\mathbf{P}_f^D/\tau)} \tag{8}$$

We tune the temperature from [0.5, 1, 2, 5] on WMT14 En-De dataset and use $\tau = 2$ as the default setting for incorporating word alignment distribution in all datasets.

- *Self-Distilled Distribution (SDD)* is the probability distribution for the source word $f$, which is produced by a same NAT model pre-trained on raw data. Specifically, the model $M$ takes a source word $f$ as input and produces a probability distribution over whole words in target vocabulary:

$$\mathbf{P}_f^M = \{P^M(e_1|f), \ldots, P^M(e_{|\mathbf{V}_{trg}|}|f)\} \tag{9}$$

This prior distribution signal can be characterized as self-distilled *lexicon level* "born-again networks" (Furlanello et al., 2018) or self-knowledge distillation (Liu et al., 2020), where the teacher and student have the same neural architecture and model size, and yet surprisingly the student is able to surpass the teacher's accuracy.

| Model | En-De | | Zh-En | | Ja-En | |
|---|---|---|---|---|---|---|
| | AoLC / LFT | BLEU | AoLC / LFT | BLEU | AoLC / LFT | BLEU |
| **AT-TEACHER** | 79.3 / 73.0 | 29.2 | 74.7 / 66.2 | 25.3 | 77.1 / 70.8 | 29.8 |
| **MaskT+KD** | 76.3 / 68.4 | 27.0 | 72.7 / 61.5 | 24.2 | 75.3 / 66.9 | 28.9 |
| **+WAD** | 77.5 / 71.9 | 27.4 | 73.4 / 64.5 | 24.8 | 76.3 / 69.0 | 29.4 |
| **+SDD** | 77.7 / 72.2 | 27.5 | 73.5 / 64.7 | 24.9 | 76.1 / 68.6 | 29.3 |
| **+Both** | 78.1 / 72.4 | 27.8 | 74.0 / 65.0 | 25.2 | 76.6 / 69.1 | 29.6 |

Table 3: Ablation Study on raw data priors across different language pairs using the MaskT Model. "WAD" denotes word alignment distribution, and "SDD" denotes self-distilled distribution. "AoLC / LFT" denotes the lexical translation accuracies for all tokens / low-frequency tokens, respectively.

| Model | Iter. | Speed | En-De | | Ro-En | |
|---|---|---|---|---|---|---|
| | | | AoLC | BLEU | AoLC | BLEU |
| **AT Models** | | | | | | |
| **Transformer-BASE** (Ro-En Teacher) | n/a | 1.0× | | 27.3 | | 34.1 |
| **Transformer-BIG** (En-De Teacher) | n/a | 0.8× | | 29.2 | | n/a |
| **Existing NAT Models** | | | | | | |
| **NAT** (Gu et al., 2018) | 1.0 | 2.4× | | 19.2 | | 31.4 |
| **Iterative NAT** (Lee et al., 2018) | 10.0 | 2.0× | | 21.6 | | 30.2 |
| **DisCo** (Kasai et al., 2020) | 4.8 | 3.2× | n/a | 26.8 | n/a | 33.3 |
| **Mask-Predict** (Ghazvininejad et al., 2019) | 10.0 | 1.5× | | 27.0 | | 33.3 |
| **Levenshtein** (Gu et al., 2019) | 2.5 | 3.5× | | 27.3 | | 33.3 |
| **Our NAT Models** | | | | | | |
| **Mask-Predict** | 10.0 | 1.5× | 76.3 | 27.0 | 79.2 | 33.3 |
|   **+Raw Data Prior** | | | 78.1 | 27.8† | 80.6 | 33.7 |
| **Levenshtein** | 2.5 | 3.5× | 77.0 | 27.2 | 79.8 | 33.2 |
|   **+Raw Data Prior** | | | 77.8 | 27.8† | 80.9 | 33.8† |

Table 4: Comparison with previous work on WMT14 En-De and WMT16 Ro-En datasets. "Iter." column indicate the average number of refined iterations. "†" indicates statistically significant difference ($p < 0.05$) from baselines according to the statistical significance test (Collins et al., 2005).

## 4.2 EXPERIMENTAL RESULTS

**Ablation Study on Raw Data Prior** Table 3 shows the results of our proposed two bilingual data dependent prior distributions across language pairs. The word alignment distribution (WAD) and self-distilled distribution (SDD) variants consistently improves performance over the vanilla *two-step training scheme* NAT model ("NAT+KD") when used individually (averagely +0.5 BLEU point), and combining them ("+Both") by simply averaging the two distributions can achieve a further improvement (averagely +0.9 BLEU point). The improvements on translation performance are due to a increase of AoLC, especially for low-frequency tokens (averagely +3.2), which reconfirms our claim. Notably, averaging the two prior distributions could rectify each other, thus leading to a further increase. We explore the complementarity of two prior schemes in Section 4.3. In the following experiments, we use the combination of WAD and SDD as the default bilingual data dependent prior.

**Comparison with Previous Work** Table 4 lists the results of previously competitive studies (Gu et al., 2018; Lee et al., 2018; Kasai et al., 2020; Ghazvininejad et al., 2019; Gu et al., 2019) on the widely-used WMT14 En-De and WMT16 Ro-En datasets. Clearly, our bilingual data-dependent prior significantly improves translation (BLEU↑) by substantially increasing the lexical choice accuracy (AoLC↑). It is worth noting that our approaches merely modify the training process, thus does not increase any latency ("Speed"), maintaining the intrinsic advantages of non-autoregressive generation.

| Frequency | En-De | Zh-En | Ja-En |
|-----------|-------|-------|-------|
| **High** | +1.3% | +0.3% | +1.3% |
| **Medium** | +0.2% | +0.1% | +0.9% |
| **Low** | **+5.9%** | **+5.8%** | **+3.3%** |
| **All** | +2.4% | +1.8% | +1.7% |

Table 7: Improvement of our approach over the MaskT+KD model on AoLC.

| Model | En-De | Zh-En | Ja-En |
|-------|-------|-------|-------|
| **NAT** | 10.3% | 6.7% | 9.4% |
| **+KD** | 7.6% | 4.2% | 6.9% |
| **+Ours** | 9.8% | 6.1% | 8.5% |

Table 8: Ratio of low-frequency target words in the MaskT model generated translations.

**Comparison with Data Manipulation Strategies** Instead of using the proposed priors, we also investigate two effective data manipulation strategies, i.e. *Data Mixing* and *Curriculum Learning*, to force the NAT model learns from both the raw and distilled data. For data mixing, we design two settings: a) Mix: simply combine the raw and distilled data, and then shuffle the mixed dataset. b) Tagged Mix: Inspired by successes of tagged back-translation (Caswell et al., 2019; Marie et al., 2020), we add tags to distinguish between KD and Raw sentences in the mixed dataset. For decay curriculum schedule, the NAT models learn more from raw data at the beginning and then learn more from KD

| Strategies | AoLC | BLEU |
|------------|------|------|
| **Baseline** | 76.3 | 27.0 |
| **Mix** | 76.6 | 27.2 |
| **Tagged Mix** | 77.1 | 27.4 |
| **Decay Curriculum** | 77.2 | 27.5 |
| **Ours** | 78.1 | 27.8 |

Table 5: Performance of several data manipulation strategies on En-De dataset. Baseline is the MaskT+KD model and Ours is our proposed approach.

as the training goes on. The details of curriculum can be found in Appendix A.3. As seen in Table 5, data mixing and decay curriculum schedule improve performance on both AoLC and BLEU, which confirm the necessity of exposing raw data to NAT models during training. Besides, our approach still outperforms those effective strategies, demonstrating the superiority of our learning scheme.

## 4.3 EXPERIMENTAL ANALYSIS

In this section, we conducted extensive analyses on the lexical choice to better understand our approach. Unless otherwise stated, results are reported on the MaskPredict models in Table 3.

**Our approach improves translation performance by reducing mis-translation errors.** The lexical choice ability of NAT models correlates to mis-translation errors, in which wrong lexicons are chosen to translate source words. To better understand whether our method alleviates the mis-translation problem, we

| Model | BLEU | AoLC | Error |
|-------|------|------|-------|
| **MaskT** | 22.6 | 68.5% | 34.3% |
| **+KD** | 24.2 | 72.7% | 30.1% |
| **+RDP** | 25.2 | 74.0% | 28.2% |

Table 6: Subjective evaluation of mis-translation errors on the Zh-En dataset.

assessed system output by human judgments. In particular, we randomly selected 50 sentences from the Zh-En testset, and manually labelled the words with lexical choice error. We defined the lexical choice error rate as $E/N$, where $E$ is the number of lexical choice errors and $N$ is the number of content words in source sentences, since such errors mainly occur in translating content words. As seen in Table 6, our approache consistently improves BLEU scores by reducing the lexical choice errors, which confirm our claim. Additionally, AoLC metric correlates well with both the automatic BLEU score and the subjective evaluation, demonstrating its reasonableness.

**Our approach significantly improves the accuracy of lexical choice for low-frequency source words.** As aforementioned discrepancy between high- & low-frquency words in Section 3.3, we focus on revealing the fine-grained lexical choice accuracy w.r.t our proposed AoLC. In Table 7, the majority of improvements is from the low-frequency words, confirming our hypothesis.

**Our approach generates translations that contain more low-frequency words.** Besides improving the lexical choice of low-frequency words, our method results in more low-frequency words being recalled in the translation. In Table 8, although KD improves the translation, it biases the NAT

model towards generating high-frequency tokens (*Low freq.↓*) while our method can not only correct this bias (averagely +32% relative change), but also enhance translation (BLEU↑ in Table 3).

**Our proposed two priors complement each other by facilitating different tokens.** As aforementioned in Table 3, combining two individual schemes can further increase the NAT performance. To explain how they complement each other, especially for low-frequency tokens, we classify low-frequency tokens into two categories according to their linguistic roles: content words (e.g. noun, verb, and adjective) and function words (e.g. preposition, determiner, and punctuation). The results are listed in Table 9. We show that WAD facilitates more on the understanding and generation of content tokens, while SDD brings more gains for function (i.e. content-free) tokens. We leave a more thorough exploration of this aspect for future work.

| Prior | AoLC on LFT | | Ratio of LFT | |
|---|---|---|---|---|
| | **Content** | **Function** | **Content** | **Function** |
| **N/A** | 67.7% | 70.1% | 5.3% | 2.4% |
| **WAD** | 71.6% | 72.9% | 5.9% | 2.5% |
| **SDD** | 71.4% | 74.3% | 5.6% | 3.4% |
| **Both** | 71.6% | 74.2% | 6.2% | 3.6% |

Table 9: AoLC and Ratio of different prior schemes on Low-Frequency Tokens ("LFT"). We list the performances on different linguistic roles, i.e. content words and function words. Note that Ratio of LFT means the ratio of low frequency tokens in generated translation. "N/A" means MaskT+KD baseline.

**Effect of Word Alignment Quality on Model Performance.** Both the proposed AoLC and priors depend heavily on the quality of word alignment, we therefore design two weaker alignment scenarios to verify the robustness of our method.

First, We adopt fast-align (Dyer et al., 2013), which is slightly weaker than GIZA++. Using fast-align, our methods can still achieve +0.6 and +0.7 improvements in terms of BLEU on En-De and Zh-En datasets, which are marginally lower than that using GIZA++ (i.e. +0.8 and +1.0 BLEU). Encouragingly, we find that the improvements in translation accuracy on low-frequency words still hold (+5.5% and +5.3% vs. +5.9% and +5.8%), which demonstrates the robustness of our approach.

In addition, we insert noises into the alignment distributions to deliberately reduce the alignment quality (Noise injection details can be found in Appendix A.4. The performances still significantly outperform the baseline, indicating that our method can tolerate alignment errors and maintain model performance to some extent.

**Effect of AT Teacher** To further dissect the different effects when applying different AT teachers, we employ three teachers. Table 10 shows our method can enhance NAT models under variety of teacher-student scenarios, including base, big and strong teacher-guided models. Our approach obtains averagely +0.7 BLEU points, potentially complementary to the majority of existing work on improving knowledge distillation for NAT models.

| AT Teacher | | NAT Model | | |
|---|---|---|---|---|
| **Model** | **BLEU** | **Vanilla** | **+Prior** | **△** |
| **Base** | 27.3 | 26.5 | 27.2 | +0.7 |
| **Big** | 28.4 | 26.8 | 27.5 | +0.7 |
| **Strong** | 29.2 | 27.0 | 27.8 | +0.8 |

Table 10: Different teachers on the En-De dataset.

## 5 RELATED WORK

**Understanding Knowledge Distillation for NAT** Knowledge distillation is a crucial early step in the training of most NAT models. Ren et al. (2020) reveal that the difficulty of NAT heavily depends on the strongness of dependency among target tokens, and knowledge distillation reduces the token dependency in target sequence and thus improves the accuracy of NAT models. In the pioneering work of NAT, Gu et al. (2018) claim that NAT suffers from the multi-modality problem (i.e. multiple lexical translations for a source word), and knowledge distillation can simplify the dataset, which is empirically validated by Zhou et al. (2020). We confirm and extend these results, showing that the AT-distilled dataset indeed leads to more deterministic predictions but propagates the low-frequency

lexical choices errors. To this end, we enhance the NAT lexical predictions by making them learn to distill knowledge from the raw data.

**Lexical Choice Problem in NMT Models**   Benefiting from continuous representations abstracted from the training data, NMT models have advanced the state of the art in the machine translation community. However, recent studies have revealed that NMT models suffer from inadequate translation (Tu et al., 2016), in which mis-translation error caused by the lexical choice problem is one main reason. For AT models, Arthur et al. (2016) alleviate this issue by integrating a count-based lexicon, and Nguyen & Chiang (2018) propose an additional lexical model, which is jointly trained with the AT model. The lexical choice problem is more serious for NAT models, since 1) the lexical choice errors (low-resource words in particular) of AT distillation will propagate to NAT models; and 2) NAT lacks target-side dependencies thus misses necessary target-side context. In this work, we alleviate this problem by solving the first challenge.

## 6    CONCLUSION

In this study, we investigated effects of KD on lexical choice in NAT. We proposed a new metric to evaluate lexical translation accuracy of NAT models, and found that 1) KD improves global lexical predictions; and 2) KD benefits the accuracy of high-frequency words but harms the low-frequency ones. There exists a discrepancy between high- and low-frequency words after adopting KD. To bridge this discrepancy, we exposed the useful information in raw data to the training of NAT models. Experiments show that our approach consistently and significantly improves translation performance across language pairs and model architectures. Extensive analyses reveal that our method reduces mistranslation errors, improves the accuracy of lexical choices for low-frequency source words, recalling more low-frequency words in the translations as well, which confirms our claim.

## 7    ACKNOWLEDGMENTS

This work was supported by Australian Research Council Projects under grants FL-170100117, DP-180103424, and IC-190100031. Xuebo and Derek were supported in part by the Science and Technology Development Fund, Macau SAR (Grant No. 0101/2019/A2), and the Multi-year Research Grant from the University of Macau (Grant No. MYRG2020-00054-FST). We also thank the anonymous reviewers for their insightful comments.

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

# A  APPENDIX

## A.1  DISCUSSION ON THE RELIABILITY OF WORD ALIGNMENT-BASED AoLC

We randomly select 20 sentence pairs from the Zh-En test set, which contains 576 source tokens. We use the trained word alignment model to produce alignments for the 20 sentence pairs, and then perform the gold word chosen procedure as described in Section 3.1. We manually evaluate these bilingual lexicons, and find that 551 out of 576 source words are aligned to reasonable equivalences (i.e. 96% accuracy). This demonstrates that it is reliable to calculate AoLC based on automatic word alignments.

## A.2  GENERAL CASES OF THE SIDE-EFFECT OF KNOWLEDGE DISTILLATION

To verify the universality of our findings that lexical choice error will propagate from teacher model, we conduct the following experiments.

In particular, we experiment AT-Base and AT-Small models on the En-De data, which are distilled by the AT-Strong model. Note that the AT-Small model consists of 256 model dimensions, 4 heads, 3 encoder and 3 decoder layers. As shown in Table 11, the same phenomena can be found in AT models when distillation is used. We leave a thorough exploration of this aspect for future work.

| Model | BLEU | AoLC on LFT | Ratio of LFT |
|---|---|---|---|
| **AT-Base** | 27.3 | 72.5% | 9.2% |
| **+KD** | 27.8 | 68.4% | 7.8% |
| **AT-Small** | 21.6 | 61.8% | 10.7% |
| **+KD** | 23.5 | 59.3% | 7.1% |

Table 11: Results of AT models on En-De when knowledge distillation is used. LFT denotes low-frequency tokens and Ratio of LFT means the ratio of low-frequency tokens in generated translation.

## A.3  DECAY CURRICULUM SETUP

Specifically, the training process is divided into 5 phases, which differ at the constituent of training data. At Phase 1, all training examples are from the raw data; and at Phase 2, 75% of the training examples are from the raw data and the other 25% are from the distilled data (note that the two kinds of training examples should cover all source sentences). Similarly, the constituent ratios at the later phases are (50%, 50%), (25%, 75%), and (0%, 100%).

## A.4  NOISE INJECTION SETUP

We swap the maximal probability tokens with other random tokens under the change ratio of N%. With 2% and 5% noises, our method respectively decreased by -0.1 and -0.2 BLEU scores on En-De. The improvements in translation accuracy on low-frequency words are +5.7% and +5.3%, which is comparable to non-noisy one (i.e. +5.9%).

