# OpenReview forum: "Understanding and Improving Lexical Choice in Non-Autoregressive Translation"
_ICLR.cc/2021/Conference — ICLR 2021 Poster_

### Official Review · AnonReviewer4 · 2020-10-27
**Official Blind Review #4**

**Rating:** 6
**Confidence:** 4

**Review:**

In *non-autoregressive* neural machine translation (NMT), learning from the predictions of *autoregressive* teacher models through sequence-level knowledge distillation (Kim and Rush, 2016) has been an important step to improve the performance of the non-autoregressive student models. Despite the success and prevalence of this knowledge distillation (KD) technique, this paper hypothesises---and empirically validates---that this KD procedure has a detrimental side-effect of propagating **lexical choice errors** that the autoregressive teacher model makes by mistranslating **low-frequency words**, into the non-autoregressive student model.

To overcome this issue, this paper proposes a way to incorporate lexical choice **prior knowledge** from the *raw parallel text* (as opposed to the autoregressive teacher's output that may propagate lexical choice errors for low-frequency words). More concretely, this work specifies two prior distributions: (i) a word alignment distribution that specifies a list of plausible target words for each token in the source sentence, as obtained from automatic word alignment tools, and (ii) a target distribution from an identical non-autoregressive teacher model trained on the raw data (i.e. *self-distillation* or born-again network), which does not contain the same lexical choice error propagation that the autoregressive teacher's model output has. The student model is then trained to minimise KL divergence with respect to these prior distributions, in conjunction with the standard sequence-knowledge distillation loss that learns from the autoregressive teacher model's output, as determined by an interpolation coefficient with logarithmic decay (Eq. 6).

**Pros:**
1. This paper does a great job of motivating the problem of lexical choice error propagation on low-frequency words from the autoregressive teacher to the non-autoregressive student. The paper clearly states its hypothesis, provides a nice motivating example, and runs extensive preliminary experiments that convincingly confirm the existence of the lexical choice problem for low-frequency words.

2.  The proposed prior knowledge approach is simple to implement, and yet provides decent improvements across all four datasets. The improvements are also consistent across different autoregressive teacher model sizes and language pairs, and the two kinds of prior knowledge can be combined to produce stronger results.

3. The paper features a fairly comprehensive analysis (including in preliminary experiments) and ablation studies that help better understand where exactly the improvements are coming from.

**Cons:**
Despite the positive aspects above, I still have two major concerns regarding the paper:
1. In Eq. 6 (page 5), the interpolation rate $\lambda$ controls how much weight is assigned to learning from the prior knowledge vs from the autoregressive teacher. But the proposed logarithmic decay function does not make sense to me. Let $i$ be the number of steps. At the very beginning of training, $i=0$, so based on Eq. 6, $\lambda = 1$. This makes sense since, at the beginning of training, the model only learns from the prior knowledge. But according to Eq. 6, $\lambda$ will actually **get larger** as training progresses (up to $i \leq I/2$). In the case where $i=I/2 - 1$, Eq. 6 will translate into $\lambda = 2$. This does not make sense for three reasons. First, $\lambda$ is an interpolation coefficient that therefore should be between 0 and 1. Second, $\lambda=2$ means that the interpolation weight assigned to distilling the autoregressive teacher is -1. Third, based on the description, $\lambda$ is designed to get smaller as more training steps are done, instead of getting larger.

2. My second concern is that there is a much simpler way of injecting the prior knowledge. For instance, what if we simply provide a decaying learning schedule (i.e. a curriculum) where, at the beginning of training, the non-autoregressive student is trained to learn more from the *raw dataset*, while at the later stages of training, the non-autoregressive student is trained to learn more from the *autoregressive teacher's output*? This can potentially accomplish the exact same goal of learning the prior knowledge from the raw dataset first, and then move on to learn more from the teacher model's output. This simpler baseline should at least be compared against.

3. This is a minor concern, but there are some grammatical mistakes and presentational suggestions in the paper that can be modified to improve clarity, as detailed below.

**Question**
1. In page 4, it is mentioned that: "The chosen procedure [to get the "gold" lexical choice for each word] is as follows: ...". How accurate is this procedure? Did you examine the output and double check whether the "gold" lexical choice corresponds well to human judgment or a dictionary entry for each word?

**Presentation/Typos/Grammatical Errors:**
1. In page 3, section 2.2, paragraph "Datasets.", "... to avoid unknown *works*..." -> "words".
2. In page 4, just under Eq. 3, it is mentioned that "$V$ is the source side vocabulary in test set". If I understand correctly, $f$ is a token on the source sentence, so shouldn't $\mathbf{V}^{test}_{src}$ be the list of *word tokens* (rather than source side vocabulary) on the source sentence?
3. In page 5, "Through fine-grained *analyzing*" -> "analysis".
4. In page 5, "...becomes significantly accurate (AoLC).." -> "significantly more accurate".
5. In page 6, "...signal of the *priority* information" -> "prior".
6. In Table 3, I would suggest displaying the AoLC performance *just on rare words* (rather than overall AoLC), since that is the problem that the paper is trying to solve.
7. In Table 5, mention what the column "Iter." means.
8. In page 7, "... It is *worthy* noting ..." -> "worth".
9.  In page 7,  "... by substantially *increase* the lexical choice accuracy ..." -> "increasing".
10. In section 4.3, I suggest saying a bit more about how the human judgment is collected.
11. In page 8, "For AT models, ..." -> remove "For".

-----Update after the authors' response-----
Thank you for the detailed authors' response, and for meticulously taking the feedback into account. The response has addressed most of my concerns.

Some further comments:

1. In Eq. 6, I think "up to $i \leq I/2$" should be replaced with "up to $i \leq (I/2 - 1)$", since $\lambda$ would be negative with when $i=I/2$. Other than this, the equation looks good to me.
2. I look forward to the addition of the results with the "decay curriculum" into the main paper. It is encouraging that the proposed approach outperforms this simpler "decay curriculum" baseline.

Since the authors have addressed most of my concerns, I am therefore raising my recommendation to a "6".

---

> ### Author Response · Authors · 2020-11-24
> **Response to AnonReviewer4**
>
> Thanks for your valuable comments and patient proofreading, which will serve to improve the paper considerably. We addressed all your concerns and questions as follows:
>
> > *[Q1] “Interpolation rate In Eq 6 will get larger as training progresses”*
>
> Sorry for the typo. In Eq. 6, $I/2(i+1)$ should be $I/(2(i+1))$, where $2(i+1)$ is the denominator.
> Accordingly, the range of the interpolation rate $log(I/(2(i+1)))/log(I/2)$ is $[0, 1]$, and the rate gradually **decreases** over training time. We have corrected the typo in the revised submission.
>
> > *[Q2] “There is a much simpler way of injecting the prior knowledge. For instance, what if we simply provide a decaying learning schedule (i.e. a curriculum) such that NAT learn more from raw at the beginning, while at the later stages learn more from kd”*
>
> This is a good point to strengthen our paper. We follow the suggestion to implement a curriculum schedule. Specifically, the training process is divided into 5 phases, which differ at the constituent of the training data. At Phase 1, all training examples are from the raw data; and at Phase 2, 75% of the training examples are from the raw data and the other 25% of the training examples are from the distilled data (the two categories of training examples cover all source sentences). Similarly, the constituent ratios at the last 3 phases are respectively (50%, 50%), (25%, 75%), and (0%, 100%). As listed in the following table, this curriculum schedule improves performance on both AoLC and BLEU, confirming our claim (the main contribution of this paper) that it is necessary to expose raw data to NAT model training. Encouragingly, our approach outperforms the decay curriculum schedule, demonstrating the superiority of our learning scheme.
>
> Models | AoLC | BLEU
> --------- | -------- | ------
> Baseline | 76.3 | 27.0
> **Decay Curriculum** | 77.2 | 27.5
> **Ours** | 78.1 | 27.8
>
> *Table 7: Performance of decay curriculum strategy on En-De dataset. The baseline is the MaskT+KD model and Ours is our proposed model.*
>
> > *[Q3] “In page 4, how accurate is the procedure [to get the gold lexical choice for each word] ? Did you examine the output and double check whether the ''gold'' lexical choice corresponds well to human judgment or dictionary entry for each word?”*
>
> We randomly select 20 sentence pairs (576 source tokens) from the Zh-En test set. We use the trained word alignment model to produce the alignment for the 20 sentence pairs, and then perform the gold word chosen procedure as described in Section 3.1. We manually evaluate these bilingual lexicons, and find that 551 out of 576 source words are aligned to reasonable equivalences (i.e. 96% accuracy). This demonstrates that it is reliable to calculate AoLC based on automatic word alignments.
>
> > *[Q4] “In page 4, just under Eq. 3, it is mentioned that "V is the source side vocabulary in the test set". If I understand correctly, f is a token on the source sentence, so shouldn't V be the list of word tokens (rather than source side vocabulary) on the source sentence?”*
>
> Your understanding is correct. We replace ''vocabulary'' with ''tokens'' in the revised version. Since the target-side sentences between KD and Raw data are different from each other, we compute the AoLC on the test set for a fair comparison.
>
> > *[Q5] “In Table 3, I would suggest displaying the AoLC performance just on rare words (rather than overall AoLC), since that is the problem that the paper is trying to solve.”*
>
> We will use additional columns to display AoLC scores on low-frequency words in the revised version. The overall AoLC scores along with the BLEU scores can demonstrate the effectiveness of our approach on improving the final performance of NAT models.
>
> We have corrected the other Presentation/Typos/Grammatical Errors listed in your comments. Thank you again for your patient and valuable proofreading.

---

### Official Review · AnonReviewer3 · 2020-11-01
**A good analysis on knowledge distillation and lexical choice in NAT**

**Rating:** 7
**Confidence:** 5

**Review:**

- Overall:

This paper analyzes the side effect of knowledge distillation in NAT where the lexical choice errors on low-frequency words are propagated to the student model from the teacher. Tackling on this, the paper then proposes to expose raw data to restore such information. In my view, the submission is well motivated and the designed experiments and results are meaningful and convincing which deserves an accept. However, as the paper focuses on analyzing a specific point (lexical choice) in a very constrained setting (NAT), the overall contribution might be incremental compared to other works in general at such a venue like ICLR.

- Specific questions:
(1) Will the same phenomena (lexical choice error propagated from teacher model) can be found in general cases when knowledge distillation is used? For instance, distillation when training small AT models. It would be nice to see a complete picture the same approach can also be used in other cases.
(2) Is the NAT model in Eq (2) the same as the final model used for translation? If no, what it looks like?
(3) What is the motivation of choosing the imitation rate $\lambda$ in Eq (6)?
(4) The improvements are still a bit marginal. It would be nice to report the statistical significance of the results.
(5) What will be the performance by simply mixing the training data with raw data instead of using the proposed prior?

---

> ### Author Response · Authors · 2020-11-24
> **Response to AnonReviewer3 (part1)**
>
> Thanks for your constructive suggestions, which will serve to improve our paper considerably. We addressed all your concerns and questions as follows:
>
> > *[Q1] “Will the same phenomena (lexical choice error propagated from teacher model) can be found in general cases when knowledge distillation is used? For instance, distillation when training small AT models. It would be nice to see a complete picture that the same approach can also be used in other cases.”*
>
> We follow your suggestion to experiment AT-Base and AT-Small models on the En-De data, which are distilled by the AT-Strong model. Note that the AT-Small model consists of  256 model dimensions, 4 heads, 3 encoder and 3 decoder layers. As shown in Table 5, the same phenomena can be found in general cases when distillation is used. We include this in the revised version and leave a thorough exploration of this aspect for future work.
>
> Model | BLEU | AoLC on LFT | Ratio of LFT
> -------- | -------- | ----------------- | ----------------
> AT-Base | 27.3 | 72.5% | 9.2%
> KD | 27.8 | 68.4% | 7.8%
> AT-Small | 21.6 | 61.8% | 10.7%
> KD | 23.5 | 59.3% | 7.1%
>
> *Table 5: Results of AT models on En-De when knowledge distillation is used. LFT denotes low-frequency tokens and Ratio of LFT means the ratio of low-frequency tokens in generated translation.*
>
> > *[Q2] “Is the NAT model in Eq (2) the same as the final model used for translation? If no, what does it look like?”*
>
> Yes, we use the final output of the NAT model to calculate the AoLC score to accurately evaluate the model behaviors. Concretely, NAT model iteratively generates the target sequence, and we use the decoding result at the final iteration.
>
> > *[Q3] “Motivation of choosing the imitation rate  in Eq (6)”*
>
> Guo et al. (2020) revealed that the logarithmic decay function performs better than the linear decay. Inspired by their findings, we adopt the logarithmic decay as the imitation rate in our work.
>
> Decaying the imitation rate allows NAT models to learn required lexical choices from the raw data by themselves. The training process is divided into two phases. At Phase 1, we only expose raw data to the NAT models at the initial time (λ=1 when i=1), and then gradually decrease the rate until 0. At Phase 2, we only expose the distilled data to NAT models, since previous studies showed that the distilled data is essential for improving the performance of NAT models. Phase 1 can be regarded as a better warm-up for Phase 2, which helps NAT models to recall lexical choices in raw data, especially for low-frequency words.
>
> [2] Junliang Guo, Xu Tan, Linli Xu, Tao Qin, Enhong Chen, Tie-Yan Liu. *Fine-Tuning by Curriculum Learning for Non-Autoregressive Neural Machine Translation*. AAAI 2020.
>
> > *[Q4] “The improvements are still a bit marginal. It would be nice to report the statistical significance of the results.”*
>
> We conduct statistical significance with sign-test (Collins et al. 2015), and show that our approach can significantly outperform strong baselines in most cases (see Table 5 in the revised version). This is encouraging since we choose strong baselines for comparison and our approach pushes the SOTA NAT performance on En-De and Ro-En benchmarks up to a better BLEU score.
>
> [3] Michael Collins, Philipp Koehn, and Ivona Kucerova. *Clause restructuring for statistical machine translation*. ACL 2005.

---

> > ### Author Response · Authors · 2020-11-24
> > **Response to AnonReviewer3 (part2)**
> >
> > > *[Q5] “What will be the performance by simply mixing the training data with raw data instead of using the proposed prior?”*
> >
> > We follow your suggestion to investigate two mixing methods:
> >
> > 1. **Mix**: we simply combine the raw and distilled data, and then shuffle the mixed dataset.
> > 2. **Tagged Mix**: Inspired by successes of tagged back-translation (Caswell et al., 2019; Marie et al., 2020), we add tags to distinguish between KD and Raw sentences in the mixed dataset.
> >
> > Models | En-De | Zh-En | Ja-En
> > ---------- | -------- | -------- | -------
> > Baseline | 27.0 | 24.2 | 28.9
> > **Mix** | 27.2 | 24.6 | 29.4
> > **Tagged Mix** | 27.4 | 24.7 | 29.3
> > **Ours** | 27.8 | 25.2 | 29.6
> >
> > *Table 6: Translation quality of NAT models using mixing strategies across different language pairs.*
> >
> > As seen, two mixing strategies can improve translation performance, which confirms the necessity of exposing raw data to NAT models during training. Besides, our approach still outperforms the simple mixing strategies, demonstrating the superiority of our learning scheme.
> >
> > In addition, we follow AnonReviewer4’s suggestion to investigate a curriculum schedule, where NAT models learn more from raw data at the beginning and then learn more from KD as the training goes on. Specifically, the training process is divided into 5 phases, which differ at the constituent of training data. At Phase 1, all training examples are from the raw data; and at Phase 2, 75% of the training examples are from the raw data and the other 25% are from the distilled data (note that the two kinds of training examples should cover all source sentences). Similarly, the constituent ratios at the later phases are (50%, 50%), (25%, 75%), and (0%, 100%). The simple curriculum schedule achieves 27.5 BLEU scores on the En-De dataset, however, our approach still performs better than this method (i.e. 27.8).
> >
> > [4] Isaac Caswell, Ciprian Chelba, David Grangier. *Tagged Back-translation*. WMT 2019.
> >
> > [5] Benjamin Marie, Raphael Rubino, Atsushi Fujita. *Tagged Back-translation Revisited: Why Does It Really Work*? ACL 2020.

---

### Official Review · AnonReviewer1 · 2020-11-02

**Rating:** 7
**Confidence:** 5

**Review:**

This paper follows up on the work (Zhou et al.) on establishing the importance of knoweldge distillation (KD) from a pretrained autoregressive translation model (AT) to train effective non-autoregresstive translation (NAT)  models. Specifically, KD is helpful because it reduces the data complexity which allows successful training of NAT models. This paper shows that KD has an undesirable effect on training of NAT models in terms of poor performance on translation into infrequent tokens and further suggests a remedy for regularizing the NAT training with an additional lexical translation loss based upon a prior translation table obtained via word alignment.

-- Two approaches for obtaining the prior distribution are explored: 1) off the shelf word alignment, 2) lexical translation probabilities based on NAT pretrained on raw data.

-- This approach is shown to be effective at improving performance of the underlying NAT model on different language pairs and the gains are consistent across multiple AT models used for KD.

-- The comparison shows that the proposed approach significantly improves translation into rare tokens compared to a vanilla NAT model trained via KD. Also, the new NAT model also tends to produce more rare tokens.

-- The proposed approach is simple to implement.

-- More ablation on the kinds of gains obtained by the two different methods for prior would have strengthened the paper. How do the distributions for low frequency words change for different language pairs under different prior schemes?

-- Moreover, both the metric used in the paper (AoLC) and the prior depend heavily on the quality of word alignment. An exploration into this aspect would have strengthened the paper.

---

> ### Author Response · Authors · 2020-11-24
> **Response to AnonReviewer1**
>
> Thanks for your positive comments and insightful suggestions, which will serve to improve the paper considerably. We addressed all your concerns, and new results were included in the revised version.
>
> > *[Q1] “How do the distributions for low-frequency tokens change for different language pairs under different prior schemes?”*
>
> We calculate the AoLC and Ratio on low-frequency tokens using different prior schemes, as shown in Table 1 and 2, respectively:
>
> Prior Scheme | En-De | Zh-En | Ja-En
> ----------------- | -------- | -------- | --------
> **WAD** | +5.2% | +5.0% | +3.1%
> **SDD** | +5.6% | +5.2% | +2.6%
> **WAD+SDD** | +5.9% | +5.8% | +3.3%
>
> *Table 1: Improvements of AoLC over the MaskT+KD on low-frequency tokens using different prior schemes.*
>
> Prior Scheme | En-De | Zh-En | Ja-En
> ----------------- | -------- | -------- | --------
> N/A	 | 7.6% | 4.2% | 6.9%
> **WAD** | 8.4% | 5.3% | 8.2%
> **SDD** | 9.1% | 5.5% | 7.6%
> **WAD+SDD** | 9.8% | 6.1% | 8.5%
>
> *Table 2: Ratio of low-frequency tokens in generated translation using different prior schemes. ''N/A'' means MaskT+KD baseline.*
>
> As seen, individual prior schemes can consistently improve both AoLC and Ratio on low-frequency tokens across different language pairs. Besides, combining two individual schemes can further increase corresponding values. To explain how they complement each other, we further classify low-frequency tokens into two categories according to their linguistic roles: **content words** (e.g. noun, verb, and adjective) and **function words** (e.g. preposition, determiner, and punctuation). The results on En-De are listed as follows:
>
> Prior Scheme | Content Tokens | Function Tokens
> ------------------ | --------------------- | ---------------------
> **WAD** | +5.8% | +4.1%
> **SDD** | +5.4% | +6.0%
> **WAD+SDD** | +5.7% | +5.9%
>
> *Table 3: Improvements in terms of AoLC of different linguistic roles over MaskT+KD on low-frequency tokens using different prior schemes.*
>
> Prior Scheme | Content Tokens | Function Tokens
> ------------------ | --------------------- | ---------------------
> N/A	 | 5.3% | 2.4%
> **WAD** | 5.9% | 2.5%
> **SDD** | 5.6% | 3.4%
> **WAD+SDD** | 6.2% | 3.6%
>
> *Table 4: Ratio of low-frequency tokens of different linguistic roles in generated translation using different prior schemes. ''N/A'' means MaskT+KD baseline.*
>
> The fine-grained analyses of different prior schemes show that WAD facilitates more on the understanding and generation of content tokens, while SDD brings more gains for function (i.e. content-free) tokens. We include the results in the revised version, and leave a more thorough exploration of this aspect for future work.
>
> > *[Q2] “Both the metric used in the paper (AoLC) and the prior depend heavily on the quality of word alignment. An exploration of this aspect would have strengthened the paper.”*
>
> Following your suggestion, we conducted additional experiments from two aspects: reliability of AoLC based on word alignment and the effect of alignment quality on model performance.
>
> 1.  **Reliability of AoLC based on Word Alignment**: We randomly select 20 sentence pairs from the Zh-En test set, which contains 576 source tokens. We use the trained word alignment model to produce alignments for the 20 sentence pairs, and then perform the gold word chosen procedure as described in Section 3.1. We manually evaluate these bilingual lexicons, and find that 551 out of 576 source words are aligned to reasonable equivalences (i.e. 96% accuracy). This demonstrates that it is reliable to calculate AoLC based on automatic word alignments.
>
> 2.  **Effect of Word Alignment Quality on Model Performance**:
>
> + We use another alignment tool fast-align, which is slightly weaker than GIZA++ used in this paper. Evaluating on the gold alignment dataset [1], the AER scores of GIZA++ and fast-align are 20.3% and 25.2% (lower AER score denotes better alignment quality). Using fast-align, our methods can still achieve +0.6 and +0.7 improvements in terms of BLEU on En-De and Zh-En datasets, which are marginally lower than that using GIZA++ (i.e. +0.8 and +1.0 BLEU). Encouragingly, we find that the improvements in translation accuracy on low-frequency words still hold (+5.5% and +5.3% vs. +5.9% and +5.8%), which demonstrates the robustness of our approach.
> + In addition, we insert noises into the alignment distributions to deliberately reduce the alignment quality (i.e. swap the maximal probability tokens with other random tokens under the change ratio of N%). With 2% and 5% noises, our method respectively decreased by -0.1 and -0.2 BLEU scores on En-De. However, the performances still significantly exceed the baseline. The improvements in translation accuracy on low-frequency words are +5.7% and +5.3%, which is comparable to non-noisy ones (i.e. +5.9%). This indicates that our method can tolerate alignment errors and maintain model performance to some extent.
>
> [1] Gold Alignment dataset is from: *http://www-i6.informatik.rwth-aachen.de/goldAlignment.*

---

### Decision · Program_Chairs · 2021-01-07
**Final Decision**

**Decision:**

Accept (Poster)

**Comment:**

This paper investigates knowledge distillation in the context of non-autoregressive machine translations. All reviewers are supportive of acceptance, especially after the thoughtful author responses. A well motivated and simple to implement approach that is giving good empirical results.